# H3K27me3 Immunohistochemical Loss Predicts Lower Response to Neo-Adjuvant Chemo-Radiotherapy in Rectal Carcinoma

**DOI:** 10.3390/biomedicines10082042

**Published:** 2022-08-21

**Authors:** Serena Ammendola, Nicolò Caldonazzi, Paola Chiara Rizzo, Giulia Turri, Corrado Pedrazzani, Valeria Barresi

**Affiliations:** 1Department of Diagnostics and Public Health, Section of Pathology, University of Verona, 37134 Verona, Italy; 2Department of Surgical Sciences, Dentistry, Gynecology and Pediatrics, Unit of General and Hepatobiliary Surgery, University of Verona, 37134 Verona, Italy

**Keywords:** rectal carcinoma, neoadjuvant chemo-radiotherapy, H3K27me3, poorly differentiated clusters, tumor budding

## Abstract

A watch-and-wait approach was suggested to avoid the possible complications related to surgery in patients with rectal carcinoma showing clinical complete response after neoadjuvant chemo-radiotherapy (CRT). Since clinical response may not correlate with pathological response, markers with higher accuracy are needed to identify patients who are likely responders and could be spared surgery. This study aims to assess whether H3K27me3 immunohistochemical expression in pre-treatment rectal carcinoma predicts response to neoadjuvant CRT or shows prognostic relevance. We assessed H3K27me3 immunostaining in 46 endoscopic biopsies of rectal carcinomas treated with neoadjuvant CRT and surgery. H3K27me3 immunostaining was lost in 20, retained in 19, and inconclusive (absent in neoplastic and non-neoplastic cells) in 7 cases. Retained H3K27me3 immuno-expression was significantly associated with ypTNM stage 0 (*p* = 0.0111) and high tumor regression, measured using either five-tiered (*p* = 0.0042) or two-tiered Dworak tumor regression grade (*p* = 0.0009). Poor differentiation, determined counting the number of poorly differentiated clusters (PDC grade) or tumor budding (TB) foci (TB grade), in the pre-treatment biopsy, was significantly associated with a shorter time to progression after surgery (*p* = 0.008; *p* = 0.0093). However, only PDC grade (*p* = 0.0023), together with radial margin involvement (*p* = 0.0001), retained prognostic significance in the multivariate analysis. The assessment of H3K27me3 immunostaining in pre-treatment endoscopic biopsy of rectal carcinoma could be useful to predict response to neo-adjuvant CRT and to identify patients who could safely undergo watch-and-wait approach. PDC and TB grade in the pre-treatment biopsy could provide additional prognostic information in patients with rectal carcinoma treated with neoadjuvant CRT and surgery.

## 1. Introduction

Colorectal carcinoma (CRC) is one of the most common malignancies worldwide and represents a major cause of cancer-related death [1]. Tumors localized at ≤ 15 cm from the anal verge are defined as rectal carcinomas and account for approximately 35% of all CRCs [2].

According to the guidelines of the European Society of Medical Oncology, the treatment of rectal carcinoma depends on its clinical Tumor Node Metastasis (cTNM) stage, established using magnetic resonance imaging (MRI) or endoscopic rectal ultrasound (ERUS) [2]. In particular, to increase tumor resectability and decrease the risk of recurrence after surgery, neo-adjuvant chemo-radiotherapy (CRT) is recommended in patients with locally advanced (cT3/4 or N+) rectal carcinoma or in those with threatened circumferential resection margin at imaging [2].

The probability of achieving a complete response to neo-adjuvant CRT depends on the initial cTNM stage and decreases in tumors involving the lower rectum or histologically showing mucinous histotype or poor differentiation [3]. Complete clinical response is defined as the lack of any palpable mass at digital rectal exam, no visible lesion at endoscopy, or as the absence of any residual tumour in the primary site and draining lymph nodes on MRI or ERUS [2]. Pathological response is measured in surgical specimen using the pathological TNM stage (ypTNM) and tumor regression grade (TRG). To assess this latter, several systems are currently in use, and they are all based on the proportion of residual tumor to stromal fibrosis in the primary tumor site [4,5,6]. In patients with complete clinical response, a “watch-and-wait” strategy could avoid the possible complications related to surgery [2]. However, clinical response may not correlate with the pathological response [7], as MRI could either underestimate (due to the limitation in discriminating between fibrosis and residual tumor) or overestimate (due to the treatment-induced tumor fragmentation) the residual tumor [2]. For this reason, other predictors of pathological tumor response could be helpful to select patients in whom a “watch-and-wait” approach could be safely applied.

Radiotherapy induces DNA double-strand breaks, and its efficacy depends and on the ability of tumor cells to repair DNA damage [8]. Not only mutations in genes encoding for proteins involved in DNA repair, but also post-translational epigenetic modifications, might influence the radio-sensitivity of tumor cells. For instance, the methylation of histone proteins, inducing chromatin condensation, might either reduce the probability of DNA damage with subsequent radio-resistance, or limit the accessibility of DNA to repair mechanisms with consequent radio-sensitivity [9]. The trimethylation status of H3 in lysine at position 27 (H3K27me3) is one of the most studied epigenetic modifications in cancer [10]. It has a proven prognostic significance in several types of central nervous system tumors, and its immunohistochemical assessment is used to classify and identify some of these entities [11]. Its effect on radio-sensitivity seems to be tumor-dependent. Indeed, the loss of H3K27me3 was associated with radio- or chemo-resistance in medulloblastomas [12], but correlated with chemo-sensitivity in ovarian carcinoma and small-cell lung cancer [9,13]. In CRC patients, the loss of H3K27me3 predicted worse outcome upon chemotherapy, and the induction of H3K27me3 enhanced oxalyplatin-mediated apoptosis in neoplastic cell lines in vitro [14]. This suggests that H3K27me3 associates with sensitivity to CRT in this tumor type.

This study aimed to assess whether the immunohistochemical expression of H3K27me3 predicts response to neo-adjuvant CRT and harbors prognostic significance in rectal carcinomas.

## 2. Materials and Methods

### 2.1. Cases

Forty-six patients with rectal adenocarcinoma, treated at our institution with neo-adjuvant CRT and surgery with total mesorectal excision (TME), were included in this study.

Inclusion criteria were: (i) pre-treatment histological diagnosis on endoscopic biopsy and availability of the corresponding paraffin block; (ii) availability of paraffin blocks or histological slides of the surgical specimen; (iii) at least 36-months follow-up.

cTNM staging was performed using digital rectal examination, total-body computerized tomography (CT), MRI, ERUS, and endoscopy with biopsy.

The clinical records were reviewed to obtain information on the tumor localization in the rectum (upper, medium, lower, or extensive), cranio-caudal extension of the tumor before treatment, tumor progression (development of metastases) during the follow-up time, and progression-free survival (PFS).

All patients were submitted to neo-adjuvant CRT based on their pre-treatment clinical stage. Standard treatment comprised radiotherapy, for a total dose of 50 Gy in 28 fractions, associated with oral Capecitabine.

Surgery was performed as per standard protocol about 6–8 weeks after CRT. The type of surgical resection was chosen depending on tumor distance from the anal verge, patients’ general conditions, and expected functional outcomes. Patients with tumors in the upper-middle rectum underwent low anterior resection with TME. If the tumor was located in the lower rectum or infiltrated the sphincter complex, ultra-low anterior resection or abdominoperineal excision (APR) were preferred.

At least 3 samples were taken for paraffin embedding from surgical specimens showing obvious tumor mass. Lesions with questionable residual tumor were entirely embedded, and if no tumor cells were detected on first paraffin sections, three additional levelled sections were examined from each paraffin block.

### 2.2. Histopathological Assessment

We revised the histological slides of the pre-treatment endoscopic biopsy, to assess the presence of poorly differentiated clusters (PDC) and tumor budding (TB) foci.

PDC were defined as clusters of at least five neoplastic cells lacking a glandular structure and counted under the microscopic field of a x20 objective lens. PDC grading was performed as previously described [15,16], and cases with no PDC, 1–2 PDC, and ≥3 PDC were classified as PDC-G1, PDC-G2, and PDC-G3, respectively. TB was quantified according to the International Tumor Budding Consensus Conference (ITBCC), and graded into Bd1 (<5 foci), Bd2 (5–9 foci), and Bd3 (>9 foci) [17].

The histological slides of the surgical specimens were reviewed to assess the ypTNM stage according to the Union for International Cancer Control (UICC, Geneve, Switzerland) (TNM 8th edition) [18], perineural invasion (PNI), lymphovascular invasion (LVI), and TRG according to Dworak [5] grading system. The assessment was performed independently by two pathologists (S.A. and V.B.); in cases of discordance, consensus was reached using a double-headed microscope. We also evaluated the status of radial (circumferential resection) margin and classified it as positive when normal tissue from the edge of the tumor measured 1 mm or less [19].

### 2.3. Immunohistochemistry

Four-µm thick sections were cut from paraffin blocks of the endoscopic biopsies and immunostained using an antibody against H3K27me3 (clone C36B11, Cell Signaling Technology, Danvers, MA, USA; dilution 1:200), by means of an automated immunostainer (Leica Biosystems, Newcastle, UK). H3K27me3 immunohistochemical expression was rated as previously described [20,21]: (i) retained, when nuclear staining was seen in ≥5% neoplastic cells; (ii) lost, when staining was absent in >95% neoplastic cells and present in internal positive controls (endothelium, stromal cells); (iii) inconclusive when staining was absent in both normal and neoplastic cells.

### 2.4. Statistical Analyses

The Chi-squared test was used to analyze the statistical correlations between H3K27me3 immuno-expression and (i) PDC and TB grades in the endoscopic biopsy, (ii) PNI, LVI, and TRG in the surgical specimen, (iii) tumor recurrence, and (iv) c- and ypTNM stage. The same test was applied to investigate the statistical correlations between tumor progression and the various clinical–pathological features (age and sex of the patients, tumor localization in the rectum, TRG, PNI, LVI, ypTNM stage, TB and PDC grades in the pre-treatment biopsy).

For the statistical analyses, we considered either five-tiered or two-tiered (TRG 0-1-2 vs. TRG 3-4) Dworak TRG.

PFS was assessed by the Kaplan–Meier method, with the date of surgery as the entry data and the length of survival to the detection of tumor progression as the endpoint. The Mantel–Cox log-rank test was applied to assess the strength of association between PFS and each of the parameters (age and sex of the patients, tumor localization and cranio-caudal extension, cTNM and ypTNM stage, infiltration of radial margin, TRG, H3K27me3 immuno-expression) as a single variable. Successively, a multivariate analysis (Cox regression model) was used to determine the independent effect of each variable on PFS.

A probability (*p*) value less than 0.05 was considered significant. Statistical analyses were performed using MedCalc 12.1.4.0 statistical software (MedCalc Software, Mariakerke, Belgium).

## 3. Results

### 3.1. Cases

The clinical–pathological features of cases in the study are resumed in Figure 1.

PDC: poorly differentiated clusters. TB: tumor budding. LVI: lympho-vascular invasion. PNI: peri-neural invasion.

Twenty-two patients were females and 24 were males (age range: 31–85 years; mean: 62 ± 12,5; median: 62 years). (2%), Eight cases (17%) were cTNM stage II and 38 (83%) were stage III.

Tumor localization was in the lower rectum in 19 (42%) cases, the medium rectum in 22 (48%), and the upper rectum in 5 (10%). The cranio-caudal extension of the tumor before treatment ranged between 2 and 9 cms (median: 4.5 cms). Twelve patients (26%) had disease progression with development of distant metastases, and three of these had local recurrence as well, during the follow-up time. PFS ranged between 18 and 86 months in patients with disease progression (median: 23 months). Follow-up time ranged between 36 and 130 months in patients without disease progression (median: 67 months).

### 3.2. Histopathological Assessment

PDC and TB could be assessed in 42/46 (91%) and 40/47 (87%) endoscopic biopsies, respectively (Figure 2).

In the other cases, the presence of a brisk inflammatory infiltrate or the extensive fragmentation prevented a reliable assessment of these features.

PDC were seen in 11/42 (26%) cases. Thirty-one tumors were classified as PDC-G1, seven as PDC-G2, and four as PDC-G3 (Figure 1).

TB foci were observed in 11/40 (28%) cases. Thirty-three cases were classified as Bd1, five as Bd2, and two as Bd3.

Complete pathological response in the rectum wall, corresponding to Dworak TRG 4, was seen in 13/46 (28%) cases.

In the surgical specimens, LVI was seen in 15 (33%) cases and PNI in 7 (15%).

Radial margin was infiltrated in 4/46 cases (8%) cases, while distal and proximal resection margins were uninvolved in all cases.

ypTNM stage was 0 in 12/46 (26%) patients, I in 11/46 (24%), II in 9/46 (20%), III in 12/47 (26%), and IV in 2/46 (4%).

### 3.3. Immunohistochemistry

H3K27me3 immunohistochemical expression was lost in 20 (43%), retained in 19 (41%), and inconclusive in 7 (15%) rectal carcinomas (Figure 3).

### 3.4. Statistical Analyses

The immunohistochemical retention of H3K27me3 in the tumor cells of the pre-treatment endoscopic biopsy was significantly associated with ypTNM stage 0 (*p* = 0.0111), and high tumor regression was measured using either five-tiered (*p* = 0.0042) or two-tiered (*p* = 0.0009) Dworak TRG (Table 1).

The probability of tumor progression during the follow-up was significantly higher in patients having a rectal cancer with Bd3 (*p* = 0.0245) or PDC-G3 (*p* = 0.0322) in the pre-treatment endoscopic biopsy, or high ypTNM stage (*p* = 0.0296), involvement of radial margin (*p* = 0.0029), LVI (*p* = 0.0287), low tumor regression, measured using two-tiered Dworak TRG, (*p* = 0.0204), in the surgical specimen (Table 2).

Tumor progression was less frequent in cases with retained H3K27me3 immuno-expression, but statistical significance was not reached (*p* = 0.0735) (Table 1).

PFS was significantly shorter in patients having a rectal cancer with Bd3 (*p* = 0.008) or PDC-G3 (*p* = 0.0093) in the pre-treatment endoscopic biopsy, high ypTNM stage (*p* = 0.0159), LVI (*p* = 0.0213) involvement of radial margin (*p* < 0.0001), or low tumor regression, measured using two-tiered Dworak TRG (*p* = 0.0336) in the surgical specimen (Table 3).

Multivariate analysis for PFS was carried out including only the variables with prognostic significance at univariate analyses. PDC-G3 (Hazard ratio: 9.5; 95% Confidence Interval: 2.2–40.9; *p* = 0.0023) and the involvement of radial margin (Hazard ratio: 14; 95% Confidence Interval: 3.7–54.7; *p* = 0.0001) (Figure 4) were the only significant and independent variables.

The PFS of patients having a rectal carcinoma PDC-G3 (*p* = 0.0093) or involving the radial margin (<0.0001) was significantly shorter than that of patients with a rectal carcinoma PDC-G1 or non-involving the radial margin

## 4. Discussion

Neo-adjuvant CRT followed by surgery with TME represents the standard of care for patients with locally advanced rectal carcinoma [2]. Although the benefit on the overall survival length is unclear [22], it decreases the rate of local recurrence and improves PFS. In the aim of avoiding the risks and long-term sequelae of surgery, watch-and-wait strategy was suggested for patients with complete clinical response after neo-adjuvant treatment [2,23]. However, 15.7% of patients managed with this approach have local regrowth within two years [23], likely because imaging overestimates tumor regression [24]. Therefore, other markers, with a higher accuracy in predicting complete pathological response, are needed to more appropriately identify patients who could be spared surgery.

Higher cTNM stage, extensive/lower involvement of the rectum, tumor size > 3 cms, TB, or mucinous histotype in the endoscopic pre-treatment biopsy have been associated with a significantly lower pathological response [3,25,26,27].

This study demonstrated that the immunohistochemical loss of H3K27me3 before treatment is associated with significantly lower pathological response (measured using ypTNM stage and Dworak TRG) of rectal carcinoma to neo-adjuvant CRT.

Indeed, 16/21 patients who had partial tumor regression (Dworak TRG 1 or 2) harbored a pre-treatment rectal carcinoma with H3K27me3 immunohistochemical loss, while 9/10 who had complete tumor regression (Dworak TRG 4) and 8/9 with ypT0N0M0 had a rectal carcinoma, which retained H3K27me3 immuno-expression.

Using tissue microarrays, two previous studies documented significantly lower expression of H3K27me3 in CRCs with advanced pTNM stage [28] or from patients with shorter disease-free survival [29], and one study found less intense immunostaining for H3K27me3 in non-irradiated rectal carcinomas of patients showing local recurrence or shorter OS [30]. These findings are consistent with a negative prognostic significance of H3K27me3 loss in CRC, similar to that reported in ependymomas [31], or meningiomas [32]. Using CRC patient-derived xenografts, one study suggested that the trimethylation of H3K27 could also be involved in the sensitivity to chemotherapy, as oxaliplatin significantly inhibited cell growth in a case with high levels of H3K27me3, but not in one with low H3K27me3 [14]. In addition, the loss of H3K27me3 was associated with a higher risk of recurrence in meningiomas treated with radiosurgery [21]. However, this is the first study to show that H3K27me3 immuno-expression could predict the response of rectal carcinoma to neo-adjuvant treatment. If confirmed in larger studies, the immunohistochemical loss of H3K27me3 in the pre-treatment endoscopic biopsy of rectal carcinoma could be used to identify patients in whom the watch-and-wait approach after neo-adjuvant CRT could not be safe because complete pathological response is unlikely. In those patients, targeting H3K27 JMJD demethylase [33] with specific inhibitors could be a potential strategy to increase radio-sensitivity, as shown in glioma cells lines [34].

In spite of its significant association with pathological response, H3K27me3 immuno-expression did not reach statistical significance as a prognostic factor on PFS after surgery. Notably, among the pathological parameters assessable on the pre-treatment endoscopic biopsy, high TB (Bd3) and PDC grades (PDC-G3) were significantly associated with a higher risk of progression and shorter PFS after surgery. The ITBCC recommends that TB is assessed at the invasive front of CRC (peri-tumoral TB) [17]. Although intra-tumoral budding (ITB) could represent a surrogate of peri-tumoral budding in CRC biopsies, its use in clinical practice is not currently recommended due to the lack of supporting evidence [35]. Two previous studies assessed TB in the biopsies of rectal carcinoma before treatment with neo-adjuvant CRT and surgery [26,27]. One used a two-tiered grading system, in which high budding was defined as the presence of at least five buds in the microscopic field of a x20 objective lens [26], while the other classified the cases as “with TB” or “without TB” [27]. Those studies demonstrated that high TB was significantly associated with lower response to neo-adjuvant CRT and shorter disease-free and cancer-specific survival [26,27]. In this study, we quantified TB according to the ITBCC recommendations [36]. Although 15% of cases could not be assessed due to extensive inflammation or tissue fragmentation, the only two tumors classified as Bd3 displayed disease progression after surgery and had significantly shorter PFS. Similar to TB, PDC reflect epithelial mesenchymal transition in CRC [37]. Herein, we confirmed findings obtained in another cohort [16] that the presence of at least two PDC in the pre-treatment biopsy predicts worse prognosis in patients with rectal carcinoma. Although a percentage of cases could not be assessed because of inflammation or tissue fragmentation, this was lower than for TB (10% vs. 15%). In addition, PDC grade, but not TB grade, was an independent prognostic variable for PFS at multivariate analysis.

In conclusion, this study shows for the first time the potential value of H3K27me3 immunostaining in predicting the response to neo-adjuvant CRT in patients with rectal carcinoma. Although we analyzed a limited number of cases, and retrospectively, if validated in other cohorts, H3K27me3 immunostaining could be performed to identify patients who could be safely spared surgery.

Confirming the prognostic value of TB and PDC in the pre-treatment endoscopic biopsy of rectal carcinomas, this study also provides support for their possible use in clinical practice in this setting.

## Figures and Tables

**Figure 1 biomedicines-10-02042-f001:**
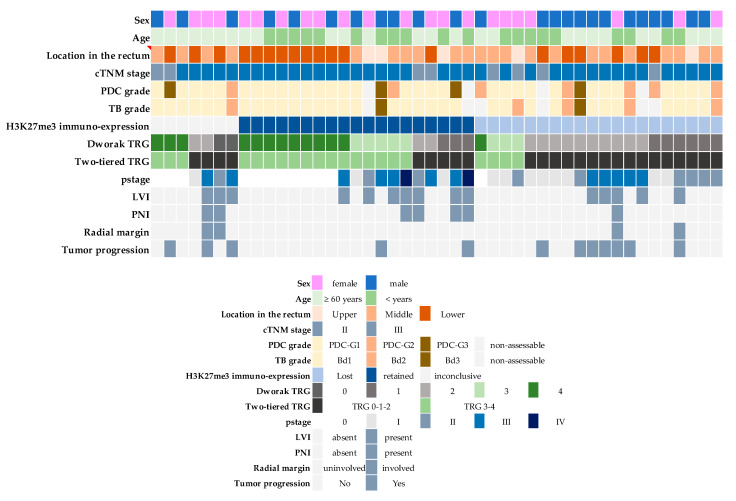
Clinical–pathological features of 46 rectal carcinomas treated with neo-adjuvant chemoradiotherapy and surgery. The loss of H3K27me3 immuno-expression in the pre-treatment endoscopic biopsy was significantly associated with lower Dworak tumor regression grade (TRG).

**Figure 2 biomedicines-10-02042-f002:**
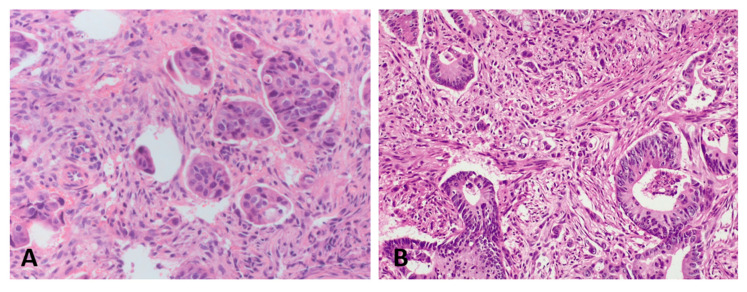
Poorly differentiated clusters (**A**) and tumor budding foci (**B**) in the pre-treatment endoscopic biopsy of rectal carcinoma.

**Figure 3 biomedicines-10-02042-f003:**
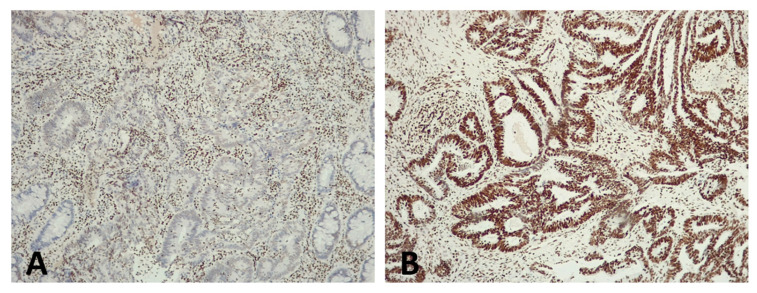
H3K27me3 immuno-expression in the pre-treatment biopsies of two rectal carcinomas. (**A**) Rectal carcinoma with loss of H3 K27me3 immunostaining in the neoplastic cells. Inflammatory cells, which served as internal positive control, showed nuclear immunostaining. (**B**) Rectal carcinoma with retained H3 K27me3 immunostaining.

**Figure 4 biomedicines-10-02042-f004:**
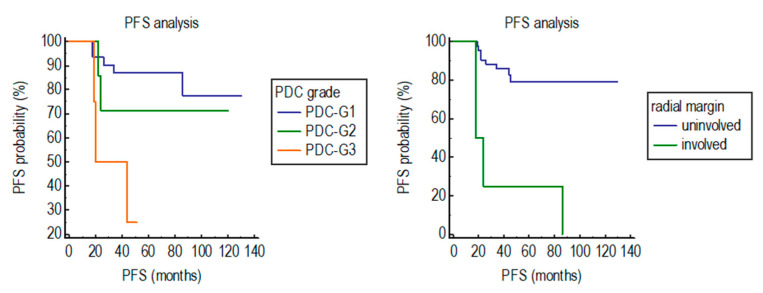
Impact of PDC grade assessed in the pre-treatment endoscopic biopsy and involvement of radial margin in the surgical specimen on the PFS of patients with rectal carcinoma treated with neoadjuvant chemo-radiotherapy and surgery.

**Table 1 biomedicines-10-02042-t001:** Statistical correlations between H3K27me3 immunoexpression and clinical-pathological features of 39 rectal adenocarcinomas treated with neoadjuvant chemo-radiotherapy and surgery.

Parameter	H3K27me3 Immuno-Expression		*p*
	Lost	Retained	
*Tumor localization*			
Lower	6	10	
Middle	12	6	
Upper	2	3	0.204
*c TNM stage*			
II	4	2	
III	16	17	0.418
*TB grade*			
Bd1	11	16	
Bd2	4	0	
Bd3	1	1	0.086
*PDC grade*			
PDC-G1	12	14	
PDC-G2	5	1	
PDC-G3	1	2	0.209
*yp stage*			
0	1	8	
I	8	2	
II	6	2	
III	5	5	
IV	0	2	0.0111
*Lymphovascular invasion*			
Absent	15	12	
Present	5	7	0.429
*Perineural invasion*			
Absent	19	15	
Present	1	4	0.139
*Dworak TRG*			
0	0	0	
1	6	3	
2	10	2	
3	3	5	
4	1	9	0.0042
*Two-tiered Dworak TRG*			
TRG 0-1-2	16	5	
TRG 3-4	4	14	0.0009
*Tumor progression*			
No	13	17	
Yes	7	2	0.0735

TB: tumor budding; PDC: poorly differentiated clusters; TRG: tumor regression grade.

**Table 2 biomedicines-10-02042-t002:** Statistical correlations between clinical–pathological parameters and tumor progression in 46 rectal adenocarcinomas treated with neoadjuvant chemo-radiotherapy and surgery.

Parameter	Tumor Progression		*p*
	No	Yes	
*Tumor localization*			
Lower	15	4	
Middle	15	7	
Upper	4	1	0.697
*c TNM stage*			
II	6	2	
III	28	10	0.939
*TB grade*			
Bd1	27	6	
Bd2	3	2	
Bd3	0	2	0.0245
*PDC grade*			
PDC-G1	26	5	
PDC-G2	5	2	
PDC-G3	1	3	0.0322
*yp stage*			
0	11	1	
I	10	1	
II	7	2	
III	5	7	
IV	1	1	0.0296
*Radial margin*			
uninvolved	34	8	
involved	0	4	0.0005
*Lymphovascular invasion*			
Absent	26	5	
Present	8	7	0.0287
*Perineural invasion*			
Absent	30	9	
Present	4	3	0.277
*Dworak TRG*			
0	1	1	
1	7	2	
2	7	7	
3	7	1	
4	12	1	0.0968
*Two-tiered Dworak TRG*			
TRG 0-1-2	15	10	
TRG 3-4	19	2	0.0204

TB: tumor budding; PDC: poorly differentiated clusters; TRG: tumor regression grade.

**Table 3 biomedicines-10-02042-t003:** Univariate analyses for PFS in 46 patients with rectal adenocarcinomas treated with neoadjuvant chemo-radiotherapy and surgery.

Parameter	H.R. (95% C.I.)	*p*
*Sex*		
Male	1	
Female	1.1 (0.3–3.6)	0.792
*Age*		
<60 years	1	
≥60 years	0.7 (0.2–2.4)	0.664
*Tumor localization*		
Lower	1	
Middle	1.5 (0.4–5.2)	
Upper	0.8 (0.1–5.7)	0.710
*TB grade*		
Bd1	1	
Bd2	2.4 (0.3–17.3)	
Bd3	8.3 (0.2–258.7)	0.008
*PDC grade*		
PDC-G1	1	
PDC-G2	1.8 (0.3–9.9)	
PDC-G3	6.9 (0.5–84.3)	0.0093
*H3K27me3 immuno-expression*		
Retained	1	
Lost	0.3 (0.09–1.3)	0.138
*c TNM stage*		
II	1	
III	1.1 (0.2–4.9)	0.859
*ypTNM stage*		
0	1	
I	0.9 (0.2–4.3)	
II	2.7 (0.5–14.7)	
III	9.1 (1.7–47.6)	
IV	6.2 (0.3–117.8)	0.0159
*Lymphovascular invasion*		
Absent	1	
Present	4.4 (1.2–15.5)	0.0213
*Perineural invasion*		
Absent	1	
Present	2.3 (0.4–11.9)	0.296
*Radial margin*		
Uninvolved	1	
Involved	318 (24.5–4127)	<0.0001
*Dworak TRG*		
0	1	
1	0.3 (0.01–9.7)	
2	0.7 (0.02–21.2)	
3	0.1 (0.005–5.6)	
4	0.1 (0.003–3)	0.139
*Two-tiered Dworak TRG*		
TRG 0-1-2	1	
TRG 3-4	0.2 (0.09–0.9)	0.0336

H.R.: hazard ratio. C.I.: confidence interval. TB: tumor budding. PDC: poorly differentiated clusters.

## Data Availability

Data will be available upon reasonable request to the corresponding author.

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
