# Peer review of "H3K27me3 Immunohistochemical Loss Predicts Lower Response to Neo-Adjuvant Chemo-Radiotherapy in Rectal Carcinoma"

_biomedicines, 2022, doi:10.3390/biomedicines10082042_

Round 1
Reviewer 1 Report
The manuscript titled "H3K27me3 immunohistochemical loss predicts lower response to neo-adjuvant chemo-radiotherapy in rectal carcinoma." is an interesting and important study demonstrating a useful method to solve a critical clinical issue.
As mentioned by the authors, if we could predict the follow-up responses to neoadjuvant chemoradiotherapy, and decide whether to carry out the surgical removal or not to prevent some complications. Hence, this study has a great scientific contribution. The overall manuscript is well-written and will interest many readers.
Three questions should be further addressed or explained.
1. How to choose the biopsy sites? The problem with biopsy is a small portion of tumors could not represent the whole tumor response. How to increase the detection reliability?
2. Is there medical imaging that could be used to detect in vivo H3K27me3 status?
3. In figure 1, the colors shown in "cTNM stage" (stage II) are different between the actual plot and the figure legend.
Author Response
REPLY TO REVIEWER-1 COMMENTS
The manuscript titled "H3K27me3 immunohistochemical loss predicts lower response to neo-adjuvant chemo-radiotherapy in rectal carcinoma." is an interesting and important study demonstrating a useful method to solve a critical clinical issue.
As mentioned by the authors, if we could predict the follow-up responses to neoadjuvant chemoradiotherapy, and decide whether to carry out the surgical removal or not to prevent some complications. Hence, this study has a great scientific contribution. The overall manuscript is well-written and will interest many readers.
Three questions should be further addressed or explained.
- How to choose the biopsy sites? The problem with biopsy is a small portion of tumors could not represent the whole tumor response. How to increase the detection reliability?
We agree with that the biopsy may not be representative of the whole tumor. However, the choice of a cut-off of >95% H3K27me3 loss, as that used in this study, likely increases the detection reliability.
- Is there medical imaging that could be used to detect in vivo H3K27me3 status?
To the best of our knowledge, no study has analyzed whether H3K27me3 status can be predicted in vivo using medical imaging.
- In figure 1, the colors shown in "cTNM stage" (stage II) are different between the actual plot and the figure legend.
Color shown in cTNM stage (stage II) has been corrected in the figure legend.

Reviewer 2 Report
This manuscript performed a retrospective analysis to investigate the relationship between the tumor progression of CRC and the levels of PTM target H3K27me3 using immunohistochemical staining. The provided data and figures are supportive to demonstrate H3K27me3 plays a vital role in the prediction of the CRC progression. Minor revision is recommended for better understanding.
1. In the introduction section, the background of H3K27me3 is relatively briefly. Some more description is suggested to explain the rationale for selecting this particular PTM target relative to others.
2. Most experimental data focused on the analysis of the patient clinical and pathological features. Only H3K27me3 immunohistochemical assessment was performed to explain the relationship between the H3K27me3 levels and CRC. Is there any additional approach to further validate this connection?
Author Response
REPLY TO REVIEWER-2 COMMENTS
This manuscript performed a retrospective analysis to investigate the relationship between the tumor progression of CRC and the levels of PTM target H3K27me3 using immunohistochemical staining. The provided data and figures are supportive to demonstrate H3K27me3 plays a vital role in the prediction of the CRC progression. Minor revision is recommended for better understanding.
- In the introduction section, the background of H3K27me3 is relatively briefly. Some more description is suggested to explain the rationale for selecting this particular PTM target relative to others.
The methylation status of H3K27me3 is one of the most studied epigenetic modifications in cancer. Due to its prognostic significance, its immunohistochemical assessment is currently used to classify and identify some tumor entities in the central nervous system. For this, it was chosen relative to other markers in this study. We have clarified this issue in the introduction section:
“The trimethylation status of H3 in lysine at position 27 (H3K27me3) is one of the most studied epigenetic modifications in cancer [10]. It has a proven prognostic significance in several types of central nervous system tumors and its immunohistochemical assessment is used to classify and identify some of these entities [11]. Its effect on radio-sensitivity seems to tumor-dependent. Indeed, the loss of…”
- Most experimental data focused on the analysis of the patient clinical and pathological features. Only H3K27me3 immunohistochemical assessment was performed to explain the relationship between the H3K27me3 levels and CRC. Is there any additional approach to further validate this connection?
The level of H3K27 methylation can be determined using other techniques, including mass spectrometry or chromatin immunoprecipitation. Although these may be more accurate to measure H3K27 methylation, immunohistochemistry was proven to be a reliable method in tumors of the central nervous system.
